# Modeling and Monitoring for Laminar Cooling Process of Hot Steel Strip Rolling with Time–Space Nature

**Qiang Wang** [1] , **Kaixiang Peng** [1,2,*] **and Jie Dong** [1]

1 Key Laboratory of Knowledge Automation for Industrial Processes of Ministry of Education, School of Automation and Electrical Engineering, University of Science and Technology Beijing, Beijing 100083, China; wqhr123@163.com (Q.W.); dongjie@ies.ustb.edu.cn (J.D.)

2 National Engineering Research Center for Advanced Rolling and Intelligent Manufacturing, University of Science and Technology Beijing, Beijing 100083, China

\* Correspondence: kaixiang@ustb.edu.cn

**Abstract:** The laminar cooling process is an important procedure in hot steel strip rolling. The spatial distribution and the drop curve of the strip temperature are crucial for the production and the quality of the steel strip. Traditionally, lumped parameter methods are often used for the modeling of the laminar cooling process, making it difficult to consider the impact of the variation of state variables and related parameters on the system, which seriously affect the stability of the steel strip quality. In this paper, a modeling and monitoring method with a time–space nature for the laminar cooling process is proposed to monitor the spatial variation of the strip temperature. Firstly, the finite-dimensional model is obtained by time–space separation to describe the temperature variation of the steel strip. Next, a global model is constructed by using the multi-modeling integration method. Then, a residual generator is designed to monitor the strip temperature where the statistics and the threshold are calculated. Finally, the superiority and reliability of the proposed method are verified by the actual-process data of the laminar cooling process for hot steel strip rolling, and different types of faults are detected successfully.

**Keywords:** process monitoring; fault detection; distributed parameter systems; time–space separation; laminar cooling process; hot steel strip rolling



## 1. Introduction

With the increasing demand for steel strips in the market, its performance is required to be higher and higher. The laminar cooling process (LCP) is an important procedure in hot steel strip rolling, which is used to cool the steel strip from an initial temperature of roughly 800–950 °C down to a coiling temperature (CT) of roughly 550–700 °C [1,2]. The CT of the steel strip after cooling process is one of the related parameters that determine the mechanical and physical performance of the steel strip. Problems or faults in the LCP, such as spray failure, side-spray system failure, and an abnormal high-level water tank, are unavoidable with the characteristics of variable operating conditions and harsh environment. If they cannot be monitored and controlled effectively, the mechanical and physical performance of the steel strip will be poor and the quality of the product will be affected eventually. Therefore, the research on modeling and process monitoring for the LCP plays a key role in improving the safety and stability of hot steel strip rolling.

The mechanism of the LCP is complex, with characteristics such as strong nonlinearity, time-varying parameters, and distributed parameters [3]. The models of the LCP commonly used for industry include the exponential model, statistical model, and differential model [4–6]. Most of these models are obtained through experiments or statistics under certain assumptions, and the accuracy is affected by the field experiment and operating conditions. In order to obtain a precise CT, a substantial body of the research has emerged, addressing the problem of modeling and monitoring for the LCP [7–13]. Many scholars

have improved the mathematical model of the LCP based on the thought of the empirical formula and the multi-model [8,9]. Cai et al. [10] accurately described the variation of the CT through the numerical calculation model, which provided an effective way for further analyzing the mechanical variation and microstructure evolution of hot steel strip rolling. Various factors affecting the CT should also be analyzed and the compensating cooling mathematical model was proposed to solve the problems of the selection of the model parameters and the large sampling period [11]. Artificial intelligence technologies such as neural networks and support vector machines are often used to optimize the accuracy of the CT for the LCP [12,13]. These models only consider the impact of the current inputs and environmental conditions on the system. Later, it has been found that the differential equations based on the mechanism of heat conduction can effectively describe the heat conduction modes in the whole cooling zone, and the distribution and the drop curve of the strip temperature can be estimated by solving the differential equations [14]. The control of the LCP is also studied for monitoring the strip temperature, for which the extended Kalman filter method was implemented to predict the CT, and the model predictive control was adopted to improve the precision of it [15]. Reducing the vibration of the process machine is also a useful method to improve the quality of the metal forming [16]. Pian et al. [17] proposed a model intelligent identification method about the temperature variation of heat conduction, which effectively improved the calculation accuracy of the coiling temperature. However, the above methods, which are called lumped parameter methods, only analyze the temperature variation of the steel strip in the time domain and ignore the influence of the variation of the boundary conditions and parameters on the system in the spatial domain. Traditional lumped parameter methods hardly describe the temperature variation in the whole cooling zone. Considering that the inputs, outputs, and even parameters can vary both temporally and spatially, and can be affected by the operating conditions, the distributed parameter systems (DPSs) are used to describe the LCP. It can accurately monitor the temperature variation of the steel strip in the spatial domain and meet the requirements of the temperature uniformity in a large spatial range.

Compared to lumped parameter systems (LPSs), the spatially distributed feature of DPSs is described by partial differential equations (PDEs), leading to the time–space nature and the degrees of freedom with infinite dimensions [18]. There has been much progress in the area of modeling in DPSs, which enriches the research of process monitoring [19–21]. Since there are a finite number of actuators and sensors for practical sensing and there is limited computing power for implementation, such infinite-dimensional systems need to be approximated by finite-dimensional systems, which is called model reduction [22]. It is obviously that the model reduction method with time–space separation becomes the main idea to solve the problem of monitoring for DPSs.

In recent years, process monitoring methods have been widely investigated, including model-based and data-driven methods [23–25]. The problem of process monitoring with infinite-dimensional properties has received significantly more attention in the existing literature. It was first considered by Demetriou, who looked at infinite-dimensional properties in space domain, and the Galerkin method was used to approximate the model to detect the fault by estimating the variation of the parameters [26]. Li et al. [27] proposed a fuzzy fault-detection filter for the hyperbolic DPSs, which were reconstructed by the T–S fuzzy model with a spatial-differential linear matrix inequality. The lumped models with a finite-dimensional order neglect the higher order; however, important modes of the system, which could lead to control or observation, spill over. To maintain the dynamic characteristic, an infinite-order observer is usually constructed for the original PDEs. Cai et al. [28] proposed a Luenberger observer to estimate the output of the system, then, the residual was proved to the convergence in the absence of disturbance, which could be reduced to a low-order one. The original infinite-dimensional data were mapped into the finite-dimensional subspace for the infinite-dimensional system to obtain a high accuracy of the residual [29]. However, these methods only depend on the mechanism analysis of the object, which is poor for analyzing the process data and being combined with them. If

the mechanism of the LCP cannot be analyzed sufficiently, it is difficult to obtain practical results with the model-based method of process monitoring.

Traditional modeling and process monitoring methods of DPSs only consider the system with a single model. Laminar cooling is an industrial process with a long procedure, in which it is difficult to describe the variation of the strip temperature with the time–space-coupled by a single model of DPSs. Based on this problem, the studies of modeling and process monitoring for the LCP need to construct a global monitoring model for data-driven realization. Firstly, a framework of modeling and process monitoring of the time–space nature is proposed for the LCP. Secondly, the finite element method is combined with the Galerkin method to describe the temperature variation in each cooling zone, and the global spatio-temporal output model for the LCP is constructed with the three-domain integration method. Finally, based on the global model, a residual generator is designed with kernel method to monitor the variation of the strip temperature when faults occur.

The rest of this paper is organized as follows: The laminar cooling process of hot steel strip rolling and its distributed parameter model is described in Section 2. In Section 3, the variation of the strip temperature in each cooling zone is analyzed and a global model is constructed to calculate the time–space output of the process. The process monitoring framework for the LCP is obtained in Section 4. In Section 5, both simulation and experiment results are presented. This article ends with concluding remarks in the last section.

## 2. Laminar Cooling Process of Hot Steel Strip Rolling

### 2.1. Process Description

The production line of hot steel strip rolling is shown in Figure 1. The hot steel strip rolling process can be divided into five steps: reheating, rough rolling, finishing rolling, laminar cooling, and coiling. In the LCP, the length of the corresponding water spray is determined by the outlet temperature, the thickness, and the speed, to adjust the total number of valves and the flow of the cooling water. More refined crystalline particles and reasonable crystal structures are obtained with better microstructures and properties in the steel strip after the cooling process. The accuracy of the CT and the distribution and the drop curve of the strip temperature have a great impact on the quality and quantity of the steel strip.

The cooling area includes the air cooling zone, the main cooling zone, and the fine cooling zone, which are affected and correlated with each other. The hot steel strip rolling process with a series structure has been formed from the product of the finish rolling coiling. The schematic diagram of the LCP is illustrated in Figure 2. The equipment of the spray headers for the LCP are installed above and below the roller table between the finishing rolling and the coiler. The top headers are of the *U*-type for laminar cooling and the bottom headers are of the straight-type for the low-pressure spray. These headers are divided into some groups in the main and fine cooling zones. Two pyrometers are located at the exit of the finishing mill and before the coiler to measure the temperature of the steel strip. The coiling speed is measured by the velocimeter installed on the coiler's mandrels. The LCP operates in the harsh environment of high-temperature baking, cooling water-vapor erosion, and vibration impact, and it is difficult to measure the variation of the temperature in the whole cooling zone. It is of great significance to establish an effective temperature monitoring model.

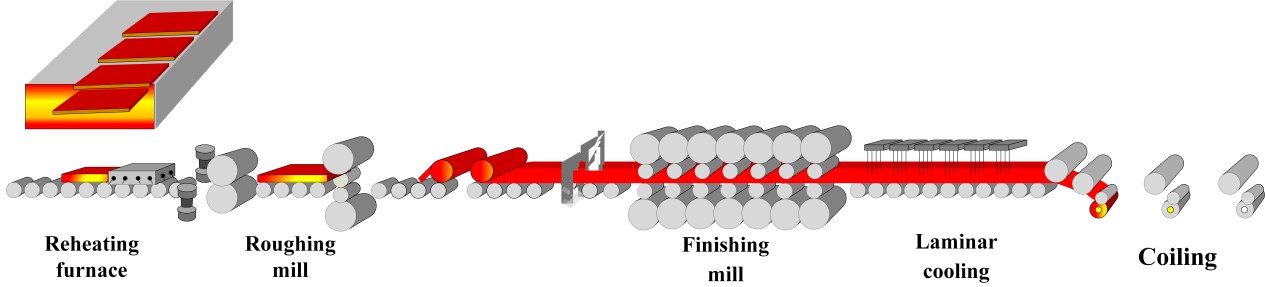

**Figure 1.** The flowchart schematic diagram of hot steel strip rolling.

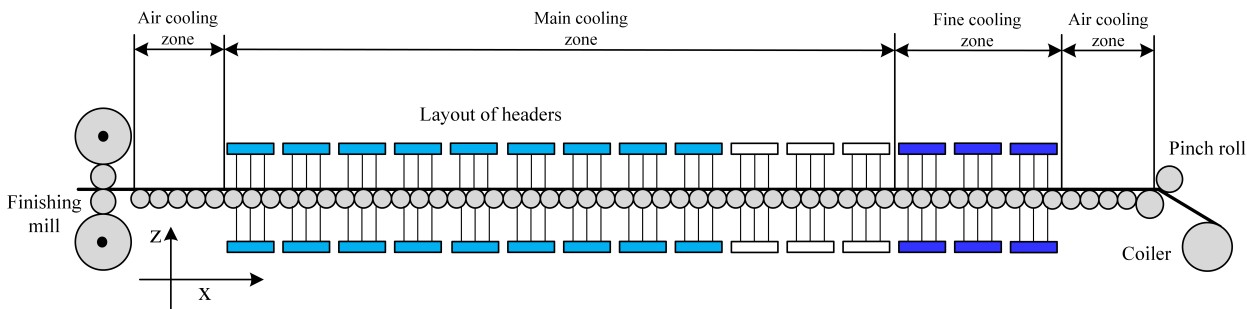

**Figure 2.** The schematic diagram of laminar cooling process.

### 2.2. Thermodynamic Model of Laminar Cooling Process

The mechanism of the LCP is usually described with the heat conduction model. The commonly used heat conduction models include the second-order parabolic PDE and second-order hyperbolic PDE, which are usually used to analyze the temperature gradient variation in the direction of the length, the width, and the thickness, with three spatial dimensions of the steel strip [30,31]. Combining the research results [32,33], a two-dimensional hyperbolic-distributed parameter model is expressed for the LCP. The PDE model is

$$\frac{\partial T}{\partial t} = -\frac{\lambda}{\rho c_p}\frac{\partial^2 T}{\partial z^2} - v \cdot \frac{\partial T}{\partial x} \tag{1}$$

where $T$ denotes the state variable; $t$ denotes the time; $x$, $z$ denote the coordinate values of the length and the thickness, respectively; $\rho$ denotes the thermal conductivity; $c_p$ denotes the specific heat capacity of the steel strip; and $\lambda$ denotes the thermal conductivity. The model assumes that the heat transfer in the width direction and length direction is considered, and the internal latent heat is ignored. The boundary conditions on its top and bottom surfaces are

$$\pm \lambda \frac{\partial T}{\partial z} = h(T - T_\infty) \tag{2}$$

$h$ depends on the heat transfer mode as follows:

$$h = h_w \frac{T - T_w}{T - T_\infty} + \sigma\varepsilon\frac{T^4 - T_\infty^4}{T - T_\infty} \tag{3}$$

where $T_\infty$ denotes the ambient temperature and $T_w$ is the water temperature; $\sigma = 5.67 \times 10^{-8}$ w/m$^2 \cdot$ K$^4$ denotes the Boltzmann constant; and $\varepsilon$ is an emission coefficient. The transfer coefficient $h_w$ is only applicable in the water cooling zone and $h_w$ is zero in air cooling zone. The boundary conditions are considered as the effect of the top and bottom headers on the temperature field, which are referred to as in reference [7].

### 3. Time–Space Coupled Based Modeling for Laminar Cooling Process

Since the steel strip runs to different cooling zones with different heat transfer conditions, the model is established to monitor the temperature variation in the whole cooling zone. A global modeling method with a time–space-coupled nature for the LCP is adopted to solve the problem. For one thing, the whole system is divided into many subsystems and each subsystem is considered by time–space separation. For another, the multi-model integration method is adopted to establish the transition relationship among subsystems to describe the whole cooling section.

#### 3.1. Local Modeling for Laminar Cooling Process

Suppose the upper and lower surfaces of the steel strip are cooled by water spraying, the temperature between surface and the core of the steel strip will be larger in a short time and a temperature gradient in the thickness direction will be formed. The average temperature in the direction of the thickness cannot be used, which will cause great error. As a result, it is difficult to effectively describe the temperature variation of the steel strip. The nodes of the steel strip, which are influenced with each other in the direction of thickness, should be considered for each cooling zone. The finite element method (FEM) constructs the model with the space-division technique, which is suitable for the LCP. With the mesh division of the steel strip, the current temperature variation can be calculated by the transfer conduction between the steel strip and water or air.

The first step of the FEM is to specify the approximate mesh for each subsystem. In the direction of the length, due to the variation of the velocity on the whole rolling line, the temperature and the velocity of the steel are always fluctuating. In order to reduce the influence of the fluctuation on the temperature field model, the steel strip is processed into blocks at the inlet and the outlet of each cooling zone in addition to the overall meshing of the steel strip. That is, the distance of the steel strip in the cooling zone of every $\Delta T$ is defined as the length of a zone of steel strip, and the variation of temperature and velocity of the steel strip can be ignored in the time domain. Each cooling zone is divided into $n_s$ grids. In the direction of the thickness, the strip is evenly divided into $2m_s$ layers to reduce the calculation error of the CT caused by the temperature gradient.

Denote the number of the grid in $x$-direction by $n_s$ and $z$-direction by $2m_s + 1$, as shown in Figure 3. $\Delta x$ and $\Delta z$ are the length and the thickness of each grid. For convenience of expression in this section, $T(z, t)$ is adopted instead of $T_i(z, t)$ to denote the state when the steel strip runs into the $i$th subsystem. $\phi_i(z)(i = 1, 2 \ldots, \infty)$ are basic functions of the state variables, which mean the weight of each node temperature in the direction of the thickness. The selection of the spatial basis function is of great importance for modeling in DPSs.

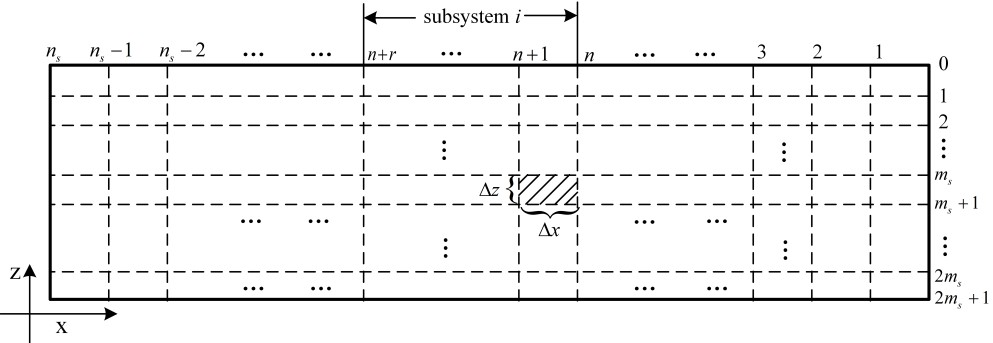

**Figure 3.** Illustration of the mesh size.

Define $f(x,t) = \frac{\partial T}{\partial x}$, which is regarded as the external inputs of the system. Then

$$\frac{\partial T(z,t)}{\partial t} = -\frac{\lambda}{\rho c_p}\frac{\partial^2 T(z,t)}{\partial z^2} - v \cdot f(x,t) \tag{4}$$

Equation (4) is called strong form. In order to obtain the solution in a numerical way, the so-called weak form is generally required with model reduction methods. Let $S$ be the function space of trial functions, which spans the exact solutions as $\phi_i(z), \phi_j(z), g(z) \in S$. To obtain the solution of the weak form, multiply both sides of (4) with $g(z)$ and integrate over $[0,1]$, then integrate by parts, which gives

$$\int_0^1 \frac{\partial T(z,t)}{\partial t}g(z)dz = -\frac{\lambda}{\rho c_p}\int_0^1 \frac{\partial^2 T(z,t)}{\partial z^2}g(z)dz - v\cdot\int_0^1 f(x,t)g(z)dz \tag{5}$$

The Galerkin method offers one possibility to approximate the exact solution in a pre-defined $m$-dimensional function subspace, in which $S$ and $S = [\phi_1(z),\phi_2(z)\ldots,\phi_m(z)]$. The spatio-temporal variable $T(z,t)$ can be expanded in $S$ as follows:

$$T(z,t) = \sum_{i=1}^m \phi_i(z)\alpha_i(t) \tag{6}$$

where $\phi_i(z)(i = 1,2\ldots,\infty)$ are basic functions of the state variables and the standard basic functions for the FEM are piece-wise polynomials, among which the first-order ones are widely used for mathematical simplicity and numerical efficiency as follows:

$$\phi_i(z) = \begin{cases} \frac{z-z_{i-1}}{z_i-z_{i-1}}, z \in [z_{i-1},z_i] \\ \frac{z-z_{i+1}}{z_i-z_{i+1}}, z \in [z_i,z_{i+1}] \end{cases} \tag{7}$$

where $i = 1,2,\ldots,\infty$. In order to reduce the numerical approximation error, either mesh size or polynomial order could be selected approximately.

Substitute (6) into (5) and replace the function $g(z)$ to the spatial term in (5) with variational basic function $\phi_j(z)$, then

$$\sum_{i=1}^n \frac{d\alpha_i(t)}{dt}\int_0^1 \Phi_i(x)g(x)dx = -\sum_{i=1}^n \alpha_i(t)\int_0^1 \frac{d\Phi_i(x)}{dx}\frac{dg(x)}{dx}dx + p\int_0^1 g(x)dx$$
$$-\frac{\lambda}{\rho c_p}\sum_{i,j=1}^m \alpha_i(t)\left\{\phi_j(1)\frac{d\phi_i(z)}{dz}|_{z=1} - \phi_i(0)\frac{d\phi_j(z)}{dz}|_{z=0}\right\} \tag{8}$$

which can be arranged as a state-space equation in the time domain:

$$\begin{bmatrix} a_{1,1} & a_{1,2} & \cdots & a_{1,m} \\ a_{2,1} & a_{2,2} & \cdots & a_{2,m} \\ \vdots & \vdots & \ddots & \vdots \\ a_{m,1} & a_{m,2} & \cdots & a_{m,m} \end{bmatrix}\begin{bmatrix} \dot{\alpha}_1(t) \\ \dot{\alpha}_2(t) \\ \vdots \\ \dot{\alpha}_m(t) \end{bmatrix}$$
$$= \begin{bmatrix} v\alpha_1(t) \\ \vdots \\ 0 \\ -v\alpha_m(t) \end{bmatrix} + \begin{bmatrix} k_{1,1} & k_{1,2} & \cdots & k_{1,m} \\ k_{2,1} & k_{2,2} & \cdots & k_{2,m} \\ \vdots & \vdots & \ddots & \vdots \\ k_{m,1} & k_{m,2} & \cdots & k_{m,m} \end{bmatrix}\begin{bmatrix} \alpha_1(t) \\ \alpha_2(t) \\ \vdots \\ \alpha_m(t) \end{bmatrix} \tag{9}$$

where $a_{i,j} = \int_0^1 \phi_i(z)\phi_j(z)dz$. Imposing the boundary condition (2) will apparently make the equations overdetermined $T(0,t) = T_{n,m}^0(t) \approx \alpha_1(t)$, then an $(m-1)$-order finite-dimensional model can be obtained:

$$
\begin{aligned}
&\begin{bmatrix} a_{2,2} & \cdots & a_{2,m} \\ \vdots & \ddots & \vdots \\ a_{m,2} & \cdots & a_{m,m} \end{bmatrix} \begin{bmatrix} \dot{\alpha}_2(t) \\ \vdots \\ \dot{\alpha}_m(t) \end{bmatrix} + \begin{bmatrix} a_{2,1}\dot{\alpha}_1(t) \\ \vdots \\ a_{m,1}\dot{\alpha}_1(t) \end{bmatrix} \\
&= \begin{bmatrix} k_{2,2} & \cdots & k_{2,m} \\ \vdots & \ddots & \vdots \\ k_{m,2} & \cdots & k_{m,m} \end{bmatrix} \begin{bmatrix} \alpha_2(t) \\ \vdots \\ \alpha_m(t) \end{bmatrix} + \begin{bmatrix} k_{2,1}\alpha_1(t) \\ \vdots \\ k_{m,1}\alpha_1(t) \end{bmatrix}
\end{aligned}
\tag{10}
$$

The right-hand side of (10) is considered as the actuator input $u(t)$, which means the number of switches, the flux of the cooling water, and the disturbance in each group of headers are as follows:

$$
u(t) = \begin{bmatrix} k_{2,1}\alpha_1(t) - a_{2,1}\dot{\alpha}_1(t) \\ \vdots \\ k_{m,1}\alpha_1(t) - a_{m,1}\dot{\alpha}_1(t) \end{bmatrix}
\tag{11}
$$

Denote $s(t) = \begin{bmatrix} \alpha_2(t) \\ \vdots \\ \alpha_m(t) \end{bmatrix}$, and the alternative representation of (10) is

$$
\dot{s}(t) = As(t) + Bu(t)
\tag{12}
$$

where $A = \begin{bmatrix} a_{2,2} & \cdots & a_{2,m} \\ \vdots & \ddots & \vdots \\ a_{m,2} & \cdots & a_{m,m} \end{bmatrix}^{-1} \begin{bmatrix} k_{2,2} & \cdots & k_{2,m} \\ \vdots & \ddots & \vdots \\ k_{m,2} & \cdots & k_{m,m} \end{bmatrix}$, $B = \begin{bmatrix} k_{2,2} & \cdots & k_{2,m} \\ \vdots & \ddots & \vdots \\ k_{m,2} & \cdots & k_{m,m} \end{bmatrix}^{-1}$. The output of the system $y(t)$ is considered as the measurement of the CT on the top and the bottom surfaces, as follows:

$$
y(t) = Es(t) + \varepsilon
\tag{13}
$$

where $E = \begin{bmatrix} \phi_2(z_1) & \cdots & \phi_m(z_1) \\ \vdots & \ddots & \vdots \\ \phi_2(z_n) & \cdots & \phi_m(z_n) \end{bmatrix}$, $\varepsilon = \begin{bmatrix} \phi_1(z_1)\alpha_1(t) \\ \vdots \\ \phi_1(z_n)\alpha_1(t) \end{bmatrix}$ with $n = 1, 2, \ldots r$ denoting $r$ sensor locations in the direction of the length for the LCP.

$y(t)$ denotes the output of the local model in each cooling zone. However, due to different production and working conditions in different cooling zones, the system often works at different multiple operating points with a long procedure. The local distributed parameter model of the LCP contains spatial functions and state variations in the time domain. It is necessary to monitor the output of the steel strip in the whole cooling zone, which requires the spatial integration method to establish the transition relationship among the cooling zones. This method will be described in detail in the next section.

### 3.2. Global Modeling for Laminar Cooling Process

To obtain a global model, direct modeling and experiments in a large operating range with strong nonlinearity and time-varying characteristics are challenging. The characteristics of the system in the spatial domain corresponding to each actuator are different in DPSs. The states varying in each cooling zone are affected by the adjacent zones. To enhance the modeling capability, spatial integration with the multi-modeling method is required to obtain a global model with a scheduling weight function at a large operating range [34]. Some studies in multi-modeling of DPSs are available in the literature, such as membership functions [35], finite Gaussian mixture models [36], kernel models [37], and collocation methods [38]. The membership function method can realize the smooth transition of the subspace and system identification of the working conditions. The output of the global model can be obtained by constructing the functional relationship between the reference point and the state in the subspace.

Each local model has different weights in different spatial locations. However, the scheduling function only depends on the time variable $t$ and the spatial variable $x$. $\mu_i(x, t)$ is regarded as the membership function in the $i$th cooling zone. To ensure the smooth transition, one way is to use traditional integration with the membership function method by reference [35]:

$$y(x,t) = \frac{\mu_1(x,t)y_1(x,t) + \cdots + \mu_i(x,t)y_i(x,t) + \cdots + \mu_{R+1}(x,t)y_{R+1}(x,t)}{\mu_1(x,t) + \cdots \mu_i(x,t) + \cdots + \mu_{R+1}(x,t)} \tag{14}$$

where $y_i(x, t)$ is the spatio-temporal output of the $i$th subsystem, $x$ denotes the distance between the $i$th subsystem and the outlet of the finish rolling, $R$ denotes the number of spray headers, and $R + 1$ denotes the air cooling zone after the water cooling zone. However, different from the traditional integration method in LPSs, the weights depend on both the system states and spatial locations of $\mu(x, s)$, where $s(x, t)$ is the state with a time–space-coupled nature of the whole system. To guarantee a smooth transition between the local model, a three-domain (3D) integration method is used to provide a global spatio-temporal model, where the weights depend on the system states (temporal variables) and spatial locations, which are shown in Figure 4. The global model is illustrated as follows:

$$y(x,t) = \frac{\mu_1(x,s)y_1(x,s) + \cdots + \mu_i(x,s)y_i(x,s) + \cdots + \mu_{R+1}(x,s)y_{R+1}(x,s)}{\mu_1(x,s) + \cdots \mu_i(x,s) + \cdots + \mu_{R+1}(x,s)} \tag{15}$$

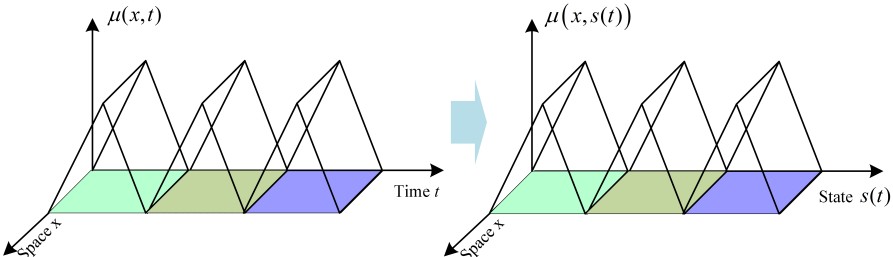

**Figure 4.** Three-domain membership functions.

Next, the functional relationship between the local model and the global one can be constructed. In the $x$-direction of the system, the global state $s(x, t)$ with a time–space-coupled nature can be calculated by the finite-dimensional method. For the global model, the output of the system $y(x, t)$ can be constructed using a set of $m$-lagged field distributions:

$$s(x,t) = [y(x,t-1), \ldots y(x,t-m)]^T \in R^d \tag{16}$$

Define the variable $\mu(s(\Omega, t))$, which can be transformed into $\mu(x, s(\Omega, t))$, and the membership function can be approximated by

$$\mu_i(x, s(x,t)) = \mu_i(x, \sum_{j=1}^{R+1} d_j(\Omega)s(t)) = w_i(x, s(t)) \tag{17}$$

The interconnection of these subsystems could be regarded as a cooperative regulation problem, and the weight functions would be calculated with the consistency analysis, as follows, by reference [39]:

$$\lim_{t \to \infty} \left\{ \|w_i - w_j\| + \left\| \frac{\partial w_i}{\partial x} - \frac{\partial w_j}{\partial x} \right\| + \left\| \frac{\partial w_i}{\partial s(t)} - \frac{\partial w_j}{\partial s(t)} \right\| \right\} = 0 \tag{18}$$

Then, the state vector is decomposed with time–space separation. $d_i(x) \in S$ has been selected as the spatial vector of the local output to obtain the relationship between the state and the membership function of the global model. Denote $\delta(x,t) = w_i(x,s(t)) - w_j(x,s(t))$, then

$$\delta(x,t) = \begin{bmatrix} \delta_2(x,t) \\ \delta_3(x,t) \\ \cdots \\ \delta_{R+1}(x,t) \end{bmatrix} = \begin{bmatrix} \sum\limits_j d_{2j}(x_2 - x_1) \\ \sum\limits_j d_{3j}(x_3 - x_2) + d_3(x_3 - x_0) \\ \cdots \\ \sum\limits_j d_{R+1,j}(x_{R+1} - x_R) + d_{R+1}(x_{R+1} - x_0) \end{bmatrix} \tag{19}$$

For each model, $w_i(x,s(t))$ can be estimated by minimizing the following cost function

$$w_i(x,s(t)) = \sum_{i=1}^{l} \sum_{j=1}^{n} \sum_{k=1}^{m} a_{ijk} \delta_j(x,t) \tag{20}$$

where $a_{i,j,k}$ are unknown parameters, which can be estimated by minimizing the following cost function:

$$\min \sum_{i=1}^{R+1} \sum_{t=1}^{L} |y_i(x,t) - \hat{y}_i(x,t)|^2 \tag{21}$$

where $y(x,t)$ is the output in each cooling zone. The solution can be obtained by many nonlinear optimization algorithms [40], and this step will minimize the global model error. $\delta_j(x,t), j = 1 \ldots R+1$ are spatio–temporal bases related to the membership function (20). After these weighting functions are obtained, the integrated global model can be used to formulate a spatio-temporal model for monitoring. $y(x,t)$ denotes the spatial variation of the CT in the direction of the length for the LCP, that is, the desired drop curve of the strip temperature into the geometrically location-dependent temperature profile from the finishing mill to the coiler.

The time–space variation of the CT for the LCP is shown in Figure 5.

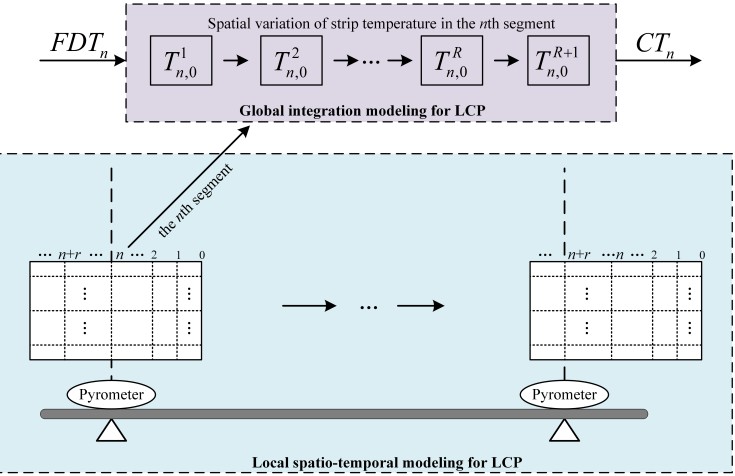

**Figure 5.** Time–space variation of the strip temperature for LCP.

## 4. Process Monitoring for Laminar Cooling Process

### 4.1. Kernel Representation-Based Residual Generator for Laminar Cooling Process

Firstly, refer to the state variables and output variables, $s(t)$ is adopted to the global model and the I/O discrete model is redefined:

$$\begin{aligned} \hat{s}(k+1) = {}&A_d(x) \cdot \hat{s}(k) + B_d(d(x), u(x,k)) \\ &+ (L(x), Y(x,k) - \hat{Y}(x,k)) \end{aligned} \tag{22}$$

$$Y(x,k) = D(x) \cdot s(k) \tag{23}$$

where $s(k) = \langle d(x), s(x,k) \rangle$, $u(k) = \langle d(x), u(x,k) \rangle$ and $\langle \cdot \rangle$ is the inner-product form. Combine (8) and (9) with (19) to calculate $A_d$ and $B_d$ with $A_d = \begin{bmatrix} 0 & I \\ -A^{-1} & -A^{-1} \cdot B \end{bmatrix} \in R^{(m+l) \cdot (m+l)}$, $B_d = \begin{bmatrix} 0 \\ A^{-1} \cdot B \end{bmatrix} \in R^{m \cdot (m+l)}$. $D(x)$ is a model parameter matrix with the spatial variable $x$, which is defined as

$$D(x) = \begin{bmatrix} d_{1,j,1}(x) & \cdots & d_{1,j,2nl}(x) \\ \vdots & \ddots & \vdots \\ d_{n,j,1}(x) & \cdots & d_{n,j,2nl}(x) \end{bmatrix}. \tag{24}$$

$L(x)$ is an appropriately chosen observer-gain matrix in $S$-space.

Define the errors $r(x,k)$ and $e(k)$ that will serve as the residual signals used for monitoring as

$$\|e(k)\| = e(x,k) = s(x,k) - \hat{s}(x,k) \tag{25}$$

$$r(x,k) = Y(x,k) - \hat{Y}(x,k) \tag{26}$$

The residual signal with a time–space-coupled nature can be transformed with kernel representation in reference [41], as follows:

$$\begin{aligned} r &= Y(x,k) - \hat{Y}(x,k) \\ &= y - D(I - A_d + LD)^{-1}(Ly + u) \\ &= (-D(I - A_d + LD)^{-1}L + I)y \\ &\quad - (D(I - A_d + LD)^{-1})u \\ &= \begin{bmatrix} -\hat{N}(x) & \hat{M}(x) \end{bmatrix} \begin{bmatrix} u \\ Y \end{bmatrix} \end{aligned} \tag{27}$$

where $\hat{M}(x) = -D(I - A_d + LD)^{-1}L + I$, $\hat{N}(x) = D(I - A_d + LD)$. The kernel $\begin{bmatrix} -\hat{N}(x) & \hat{M}(x) \end{bmatrix}$ represents the redundancy of the DPSs. Different from the process monitoring method in the LPS, the input signal obtained in the global model is infinite-dimensional.

Substitute the spatial information of the local model into the residual and define $Y_{k,k+s}$ and $u_{k,k+s}$, which are related to $y(x,k)$ and $u(k)$, then

$$Y_{k,k+s}(x,k) = \begin{bmatrix} (d(x), y(x_1,k)) & \cdots & (d(x), y(x_n,k+N-1)) \\ \vdots & \ddots & \vdots \\ (d(x), y(x_1,k+s)) & \cdots & (d(x), y(x_n,k+s+N-1)) \end{bmatrix}$$

$$u_{k,k+s}(x,k) = \begin{bmatrix} (d(x), u(x_1,k)) & \cdots & (d(x), u(x_n,k)) \\ \vdots & \ddots & \vdots \\ (d(x), u(x_1,k+s)) & \cdots & (d(x), u(x_n,k+s)) \end{bmatrix} \tag{28}$$

the I/O model can be formulated as

$$Y_{k,k+s} = \Gamma_s \cdot \Phi_s + H_{u,s} \cdot u_{k,k+s} \tag{29}$$

where $\Gamma_s$ and $H_{u,s}$ are the transfer matrices after the expansion of the input and output, as follows:

$$\Gamma_s = \begin{bmatrix} D \\ D \cdot A_d \\ \vdots \\ D \cdot A_d^s \end{bmatrix}, \Phi_s = \begin{bmatrix} s_0(k) \\ s_1(k) \\ \vdots \\ s_{s+1}(k) \end{bmatrix},$$

$$H_{u,s} = \begin{bmatrix} I & 0 & 0 & 0 \\ D \cdot L & \ddots & \ddots & 0 \\ \vdots & \ddots & \ddots & 0 \\ DA_d^{s+1} \cdot L & \cdots & D \cdot L & I \end{bmatrix} \tag{30}$$

It follows from (27) that the core of the residual generation problem is to identify the left coprime factorization. This model is obtained in the data-driven realization, which is referred to as in reference [42]. This is an alternative way of designing a residual generator directly. This motivates us to address the data-driven design of the process monitoring problem, which can be schematically formulated as an equation described by

$$\forall u_{k,k+s} \psi \cdot \begin{bmatrix} u_{k,k+s} \\ Y_{k,k+s} \end{bmatrix} = 0 \tag{31}$$

where $\psi$ is the data-driven realization of the left coprime factorization, which should be calculated for the residual generator.

Assume that $\psi = \begin{bmatrix} \psi_{s,u} & \psi_{s,y} \end{bmatrix}$. Since the I/O model of (28) cannot be identified and applied to the residual generation, QR decomposition in reference [43] is used to estimate the influence on the residual, which is corresponding to the space vector, and then

$$\begin{bmatrix} \Phi_s \\ u_{k,k+s} \\ Y_{k,k+s} \end{bmatrix} = \begin{bmatrix} P_{11} & 0 & 0 \\ P_{21} & P_{22} & 0 \\ P_{31} & P_{32} & P_{33} \end{bmatrix} \begin{bmatrix} Q_1^T \\ Q_2^T \\ Q_3^T \end{bmatrix} \tag{32}$$

Moreover,

$$H_{r,s} \cdot R_{k,k+s}(Q_3 \cdot Q_3^T) = L_{33} \cdot Q_3^T \tag{33}$$

and the useful information about the lumped residual generator is mainly included in $\begin{bmatrix} P_{21} & P_{22} \\ P_{31} & P_{32} \end{bmatrix}$. By doing the following SVD:

$$\begin{bmatrix} P_{21} & P_{22} \\ P_{31} & P_{32} \end{bmatrix} = \begin{bmatrix} U_1 & U_1 \end{bmatrix} \begin{bmatrix} \Sigma_1 & 0 \\ 0 & \Sigma_2 \end{bmatrix} \begin{bmatrix} Q_2^T \\ Q_3^T \end{bmatrix} \tag{34}$$

$\psi$ and $\begin{bmatrix} P_{21} & P_{22} \\ P_{31} & P_{32} \end{bmatrix}$ have the same left null matrix, then

$$\Sigma_2 = 0, \psi_s = U_2^T \tag{35}$$

The equation can be proved by reference [44], then:

$$\psi_{s,y} \cdot \Gamma_s = 0, \psi_{s,u} = -\psi_{s,y} \cdot H_{u,s} \tag{36}$$

and we obtain the coprime factorization $\psi = \begin{bmatrix} \psi_{s,u} & \psi_{s,y} \end{bmatrix}$.

Then, calculate the observer gain $L(x)$. Since $L(x)$ is a matrix whose elements are the functions of $x$, $L(x), d(x) \in S$, for convenience, let us assume that $L(x)$ can be designed based on $d(x)$, which is defined according to the properties of basic functions, as follows:

$$L_{i,j}(x) = \sum_{k=1}^{\gamma} \eta_{i,j,k} d_k(x), i = 1, \dots 2l, j = 1, \dots l \tag{37}$$

Assume that $(L(x), D(x)) = L \cdot D$, where $L_i = [L_{i,1}, \dots, L_{i,n}]$, then

$$L_{i,j} = \begin{bmatrix} l_{1,j,1} & \cdots & l_{1,j,l} \\ \vdots & \ddots & \vdots \\ l_{2n,j,1} & \cdots & l_{2n,j,l} \end{bmatrix} \tag{38}$$

It can be noted that $L(x)$ contains the weighting coefficients for (38). Substitute the result of $D(x)$ into (37) with

$$\begin{aligned} L(x) &= \begin{bmatrix} L_{i,1}\hat{d}(x) & \cdots & L_{i,n}\hat{d}(x) \end{bmatrix} \\ &= \begin{bmatrix} l_{1,j,1} & \cdots & l_{1,j,l} \\ \vdots & \ddots & \vdots \\ l_{2n,j,1} & \cdots & l_{2n,j,l} \end{bmatrix} \cdot \begin{bmatrix} (d_1(x), d_{n,j,1}(x)) & \cdots & (d_1(x), d_{n,j,2nl}(x)) \\ \vdots & \ddots & \vdots \\ (d_l(x), d_{n,j,1}(x)) & \cdots & (d_l(x), d_{n,j,2nl}(x)) \end{bmatrix} \end{aligned} \tag{39}$$

where $L(x)$ should be selected in such a way that $(A_d, (L(x), D(x)))$ is stable and the observer gain can be calculated by this method. Based on them, the residual generator is achieved:

$$r(x, k) = \begin{bmatrix} \psi_{s,u} & \psi_{s,y} \end{bmatrix} \cdot \begin{bmatrix} u_{k,k+s} \\ Y_{k,k+s} \end{bmatrix} \tag{40}$$

*4.2. Residual Evaluation and Threshold Setting*

The Hotelling's $T^2$ test statistic is adopted to evaluate the residual, and $T^2$ is defined as

$$T^2 = r^2/\sigma_r \tag{41}$$

where $\sigma_r$ is the covariance matrix of the residual signal when there is no fault with

$$\sigma_r = \frac{1}{N-1} \sum_{k=1}^{N} \left( r - \frac{1}{N} \sum_{k=1}^{N} r \right)^2 \tag{42}$$

where $N$ is the length of evaluation window. Under normal operating conditions, we have $r(k) \sim N(r_l, \Sigma)$. As a result, the $T^2$ test statistic is noncentrally $\chi^2$-distributed with $m$ degrees of freedom, and the noncentrally parameter is $r_f/\sigma_r^2$. The distribution of the test statistic is

$$T^2 \sim \chi^2(1, r^2/\sigma_r) \tag{43}$$

For a given significance level $\alpha$, the threshold is determined as

$$J_{th,T^2} = \chi_{1-\alpha}^2(1, r_f^2/\sigma_r) \tag{44}$$

and the decision logic is as follows:

$$\begin{cases} T^2 > J_{th,T^2}, faulty \\ otherwise, fault\ free \end{cases} \tag{45}$$

The schematic procedure of the proposed process monitoring approach is presented in Figure 6.

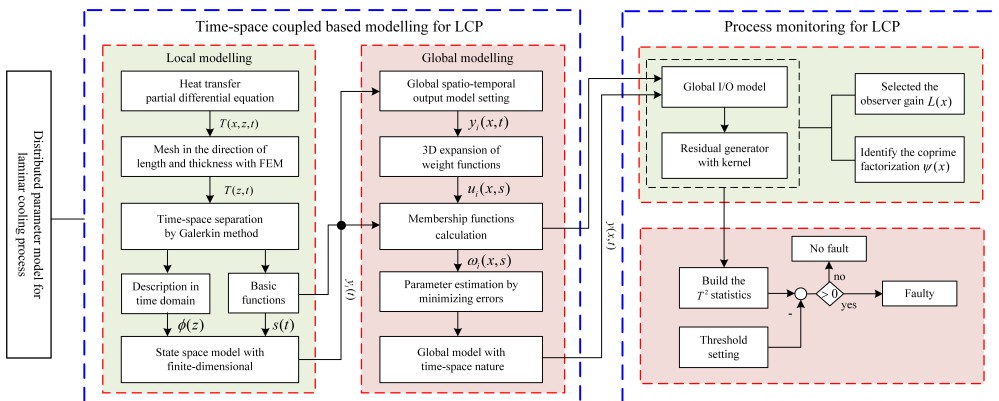

**Figure 6.** Flowchart of the proposed modeling and process monitoring method for LCP.

## 5. Case Studies

In this section, a simulation study is carried out to verify the reliability and effectiveness of the proposed method in modeling and monitoring for the LCP. Carbon Q235B-type steel is taken as an example and the data are the actual operation data of a steel plant. The parameters of the Q235B steel strip are shown in Table 1.

**Table 1.** Q235b strip steel thermophysical parameters.

| Item | Value | Unit |
|:---:|:---:|:---:|
| $\lambda$ | $\begin{cases} 42.5, T \in [400, 600) \,^{\circ}\text{C} \\ 36.3, T \in [600, 800) \,^{\circ}\text{C} \\ 29.6, T \in [800, 1000) \,^{\circ}\text{C} \end{cases}$ | $\text{w} \cdot (\text{m} \cdot {}^{\circ}\text{C})^{-1}$ |
| $c_p$ | $\begin{cases} 621.2, T \in [400, 600){}^{\circ}\text{C} \\ 716.4, T \in [600, 800){}^{\circ}\text{C} \\ 946.7, T \in [800, 1000){}^{\circ}\text{C} \end{cases}$ | $\text{J} \cdot (\text{kg} \cdot {}^{\circ}\text{C})^{-1}$ |
| $\rho$ | 7850 | $\text{kg} \cdot \text{m}^3$ |
| $T_w$ | 30 | ${}^{\circ}\text{C}$ |
| $T_\infty$ | 30 | ${}^{\circ}\text{C}$ |

### 5.1. Validation of the Model for Laminar Cooling Process

The laminar cooling process studied in this paper is illustrated in Figure 2. There are 14 header banks in laminar cooling equipment as $l_1$–$l_{14}$, which all include top and bottom headers. The initial state of the spray headers is shown in Figure 7. $l_1$–$l_9$ are open with the water flux of $185 \, \text{m}^3 \cdot (\text{s} \cdot \text{m}^2)^{-1}$ and the top and bottom surfaces of the steel strip both transfer heat with the cooling water, which is called top and bottom water cooling. $l_{10}$–$l_{11}$ are closed, and the top and bottom surface of the steel strip both transfer heat with air, which is called top and bottom air cooling. The last three banks are regarded as the fine cooling zone, with a water flux of $120 \, \text{m}^3 \cdot (\text{s} \cdot \text{m}^2)^{-1}$, and the state of the $l_{14}$ header is set to be open in the top and closed in the bottom.

The strip data of actual production are applied to validate the global model for the LCP. Five groups of measured steel strip data are selected, and each group of the strip data includes 160 segments of input and output. The steel strip is divided into several zones with a period of 1 s. The control system gives the set value of the number of the total opening in a cycle of 1 s. When the head of the steel strip enters the cooling zone, the control period is consistent with the time step. The inlet of the cooling zone has monitored the temperature and speed of the steel strip in a period of 1 s, and the outlet detected the coiling temperature of the steel strip, also. As for the method of model reduction, the open dynamic system mentioned above is divided into eight segments in the $z$-direction and 16 segments in the $x$-direction.

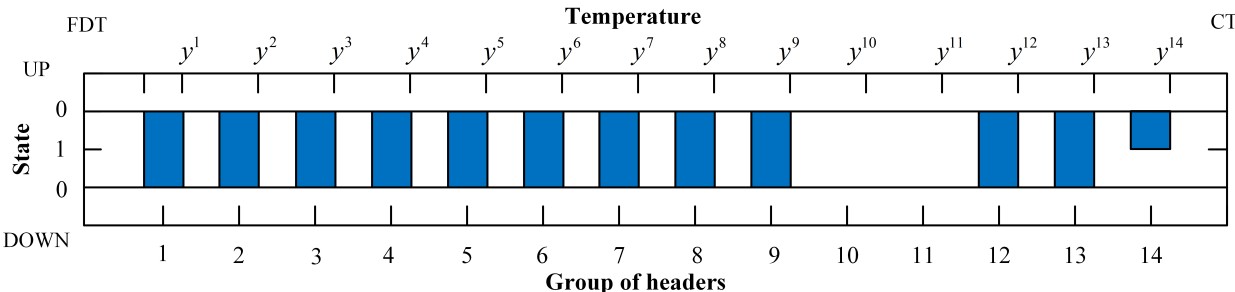

**Figure 7.** The initial state of spray header banks in LCP.

Based on the same specifications and input conditions, the global model of the LCP is established in Section 3, and is used to calculate the CT, and the accuracy of it is compared with the measured CT. The data with same specifications are shown in Table 2, which are selected from the same steel strip. $N_d$ and $N_s$ denote the sequence number of all and each group of the data. $U_{main}$ and $U_{fine}$ are the number of headers open in the main cooling zone and the fine cooling zone. $G$ is the strip thickness of finishing rolling. $FRT_0$ and $v_0$ are finishing rolling temperature and velocity of the strip heads, respectively. $FRT_n$, $v_n$, and $a_n$ are the measured data for finishing rolling temperature, velocity, and acceleration of the $n$th segment at the exit of finishing mill, respectively. The outputs of the data include the coiling temperature $CT_n$ and coiling velocity $\hat{v}_n$ at the cooling zone outlet.

**Table 2.** Input and output data of modeling with same specifications for LCP.

| $N_d$ | $N_s$ | $U_{main}$ | $U_{fine}$ | $G$ | $FDT_0$ (°C) | $v_0$ (m/s) | $FDT_n$ (°C) | $v_n$ (m/s) | $a_n$ (m/s²) | $CT_n$ (°C) | $CT$ (°C) |
|---|---|---|---|---|---|---|---|---|---|---|---|
| 1 | 0 | 108 | 60 | 5.25 | 872 | 4.00 | 872 | 4.00 | 0.5 | 658 | 650 |
| 2 | 1 | 108 | 60 | 5.25 | 872 | 4.00 | 864 | 4.04 | 0.5 | 655 | 650 |
| 3 | 2 | 108 | 60 | 5.25 | 872 | 4.00 | 863 | 4.09 | 0.5 | 652 | 650 |
| 4 | 3 | 108 | 60 | 5.25 | 872 | 4.00 | 866 | 4.15 | 0.5 | 649 | 650 |
| ... | ... | ... | ... | ... | ... | ... | ... | ... | ... | ... | ... |
| 159 | 158 | 108 | 60 | 5.25 | 872 | 4.00 | 870 | 4.17 | −0.4 | 658 | 650 |
| 160 | 159 | 108 | 60 | 5.25 | 872 | 4.00 | 871 | 4.15 | −0.4 | 656 | 650 |

The resulting predictions and the measurements of the CT are shown in Figure 8. By comparison and calculation of the results, the curve of the predictive CT is very close to that of the measurement. The maximum error between the prediction and the measurements is nearly 13 °C. There are 34, 98, and 28 sampling points with errors within 5 °C, 5–10 °C, and 10–20 °C, respectively, as a result of the accumulation of errors in each cooling zone. The prediction curve of the coiling temperature is smoother than the measurement curve with the increase of the sampling points.

Based on the global model with a time–space coupled nature, the variation of the strip temperature with a variable segment in each cooling zone can be observed. In order to further validate the designed model, the 1st cooling zone (headers are all open, located at the beginning of the main cooling zone), the 10th cooling zone (headers are all closed, located at the back end of the main cooling zone), and the 14th cooling zone (headers are partially open, located at the exit of the fine cooling zone) are selected to observe the spatial variation of the strip temperature from zero, one, and four nodes in the direction of the thickness in each cooling zone.

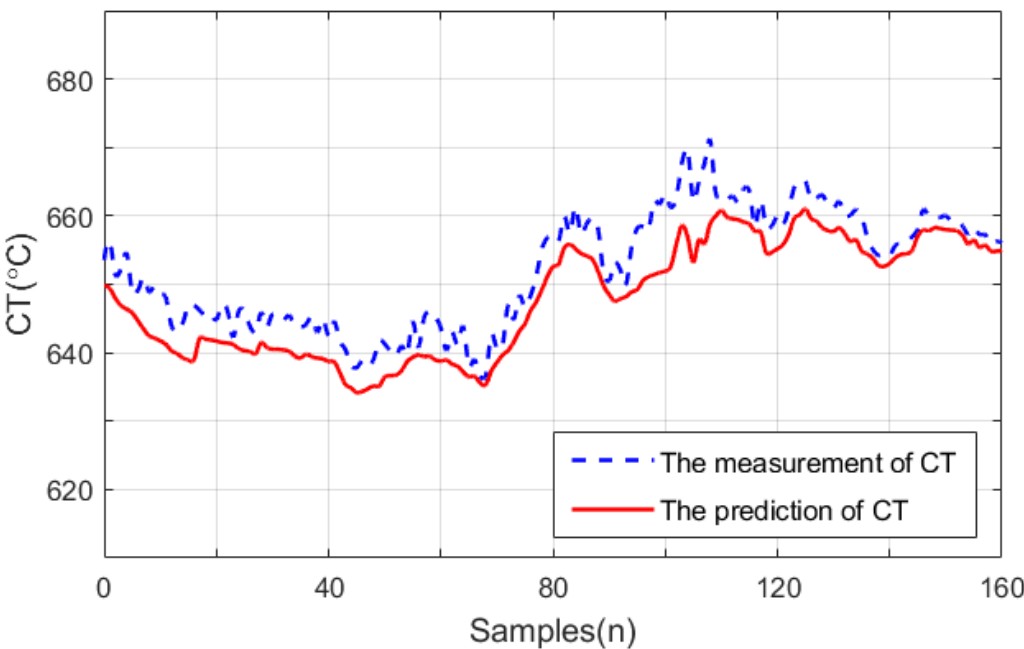

**Figure 8.** Comparison between the prediction and the measurement of CT with same specifications.

Simulations are performed to illustrate this in Figure 9a–c. The top-surface temperature of the steel strip is lower than the middle node of the steel strip due to the heat transfer in the direction of the thickness. Besides, when the strip enters the cooling zone with a different transfer conduction, the temperature variation is also different in the direction of the thickness. The steel strip runs to the 14th cooling zone and the temperature variation in the thickness direction is more obvious than that in other cooling zones. The trend of the temperature variation in each cooling zone is similar to the prediction of the CT, which is also reflected in the temperature gradient in the direction of the thickness.

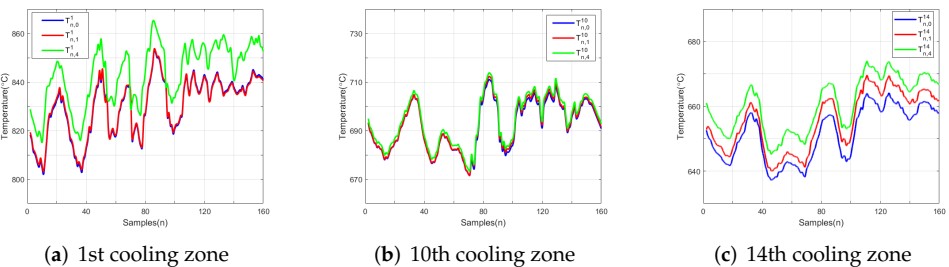

(**a**) 1st cooling zone      (**b**) 10th cooling zone      (**c**) 14th cooling zone

**Figure 9.** Spatial variation of the strip temperature in the 1st (**a**), the 10th (**b**), and the 14th (**c**) cooling zones.

The spatio-temporal output is illustrated in Figure 10. The three-domain coordinates represent the segment number, each cooling zone, and the temperature of the steel strip. "Temperature-samples" in two-dimensional space denotes the spatial variation in each cooling zone of any node. "Temperature-positions" means the temperature drop curve of the strip into the geometrically location-dependent temperature profile from finishing mill to coiler.

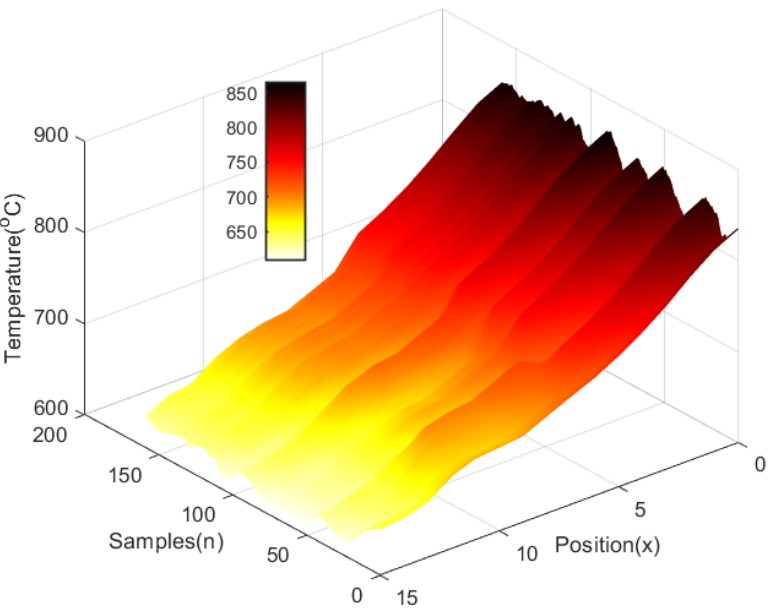

**Figure 10.** The spatial distribution of the strip temperature in normal condition.

The verification above is carried out in the same specification. In order to validate the model with a different specification, data from four steel strips are used for the experiment ($G$, $FDT_n$, and $CT_n$ are different), as seen in Table 3. The comparison between the prediction and the measurement of the CT for different specifications under the same input conditions is illustrated in Figure 11. It can be seen that even if the specifications of the steel strip change, the CT calculated by the dynamic model basically fits the measured coiling temperatures with small errors. In conclusion, the global model proposed in this paper is an effective method for the LCP, which could describe the spatial variation of the temperature in the whole cooling zone.

**Table 3.** Input and output data of modeling with different specifications for LCP.

| $N_d$ | $N_s$ | $U_{main}$ | $U_{fine}$ | $G$ | $FDT_0$ (°C) | $v_0$ (m/s) | $FDT_n$ (°C) | $v_n$ (m/s) | $a_n$ (m/s²) | $CT_n$ (°C) | $CT$ (°C) |
|---|---|---|---|---|---|---|---|---|---|---|---|
| 1 | 0 | 96 | 60 | 5.25 | 872 | 4.00 | 865 | 4.18 | 0.5 | 658 | 650 |
| 2 | 1 | 96 | 60 | 5.25 | 872 | 4.00 | 868 | 4.23 | 0.5 | 655 | 650 |
| 3 | 2 | 96 | 60 | 5.25 | 872 | 4.00 | 872 | 4.26 | 0.5 | 652 | 650 |
| 4 | 3 | 96 | 60 | 5.25 | 872 | 4.00 | 873 | 4.29 | 0.5 | 649 | 650 |
| ... | ... | ... | ... | ... | ... | ... | ... | ... | ... | ... | ... |
| 159 | 158 | 96 | 60 | 5.25 | 872 | 4.00 | 868 | 4.62 | −0.4 | 658 | 650 |
| 160 | 159 | 96 | 60 | 5.25 | 872 | 4.00 | 864 | 4.59 | −0.4 | 656 | 650 |
| 161 | 0 | 96 | 60 | 5.25 | 872 | 4.00 | 856 | 4.64 | 0.5 | 645 | 650 |
| 162 | 1 | 96 | 60 | 5.25 | 872 | 4.00 | 853 | 4.67 | 0.5 | 646 | 650 |
| ... | ... | ... | ... | ... | ... | ... | ... | ... | ... | ... | ... |
| 481 | 0 | 108 | 60 | 5.25 | 872 | 4.00 | 858 | 4.61 | 0.5 | 641 | 650 |
| ... | ... | ... | ... | ... | ... | ... | ... | ... | ... | ... | ... |
| 639 | 158 | 108 | 60 | 5.25 | 872 | 4.00 | 875 | 4.24 | −0.4 | 659 | 650 |
| 640 | 159 | 108 | 60 | 5.25 | 872 | 4.00 | 873 | 4.21 | −0.4 | 658 | 650 |

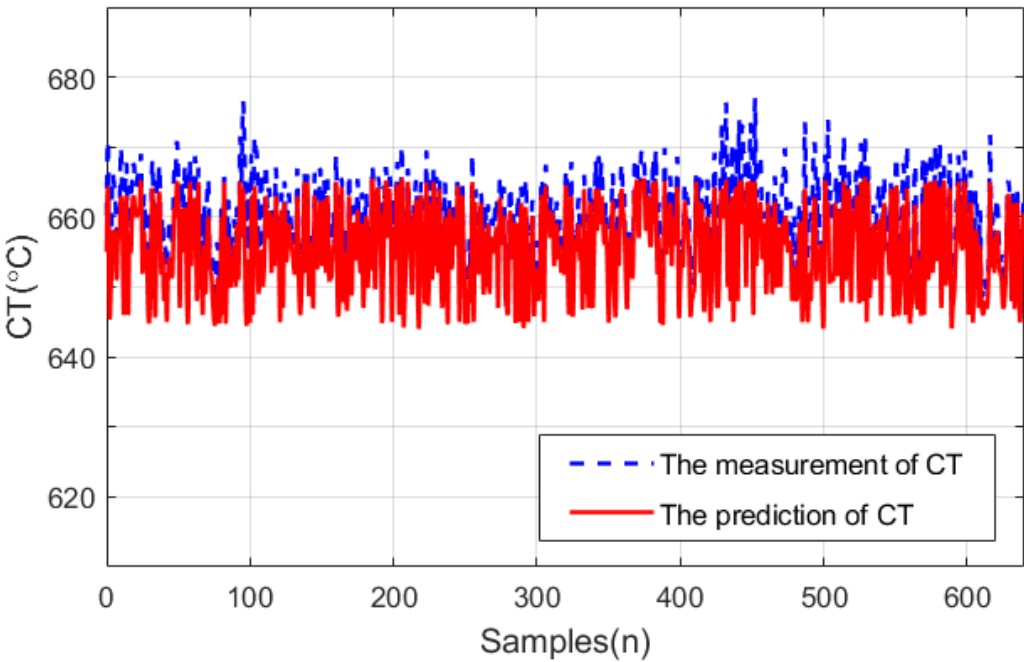

**Figure 11.** Comparison between the prediction and the measurement of CT with different specifications.

### 5.2. Performance of Process Monitoring

Based on the constructed residual generators, the abnormal conditions are used to verify the effectiveness of the proposed approach. Faults in the LCP include spray failure, instrument failure, side-spray system failure, and high-level water tank failure. In this section, the spray failure and the instrument failure are selected to observe the spatial variation of the strip temperature when fault occurs. Three fault scenarios are considered as follows:

- The CT of the LCP is going up due to the failure of the 14th spray headers in the fine cooling zone (the switch of the bottom headers is blocked and the headers fail to close)(105th sample);
- The CT of the LCP is decreased due to the failure of the ninth spray headers in the main cooling zone (the main switch of the group is blocked and the headers fail to be closed)(80th sample);
- The CT of the LCP is varied in many samples due to the failure of speed tachometers for measuring coiling speed, resulting in the running speed of the steel strip slowing down at subsequent nodes (90th sample).

The spatial variations of the steel strips of different faults are studied with the same input and working conditions. The first 160 samples are used to construct the test statistics and to determine the threshold during the offline phase. With this fault structure, the observer gain $L(x)$ could be designed in such a way that eigenvalues of $(A_d - (L(x), c(x)))$ are all 0.01 under normal conditions. Note that the identified $L(x)$ is not provided to the online scheme and is only used to validate the performance of the scheme.

Similar to the simulation studies, we initialized the observers with the same specification conditions, except for the state of spray headers. A total of 640 samples of data with four steel strips are used for the calculation of the threshold in normal conditions, and five basis functions of piece-wise polynomials are used to construct the residual generator during the offline phase. Therefore, we evaluate the effectiveness of the proposed scheme in terms of the residual under faulty conditions. The surface temperatures under faulty cases are shown in Figures 12a, 13a, and 14a, which clearly show the spatial variation of the temperature under faulty conditions. The threshold setting is 9.4525. The detection results of three fault scenarios are given in Figures 12b, 13b, and 14b. It can be observed

that the temperature distribution with an obvious change happened in faulty, compared to normal, conditions. From the 105th, 80th, and 90th samples, all three kinds of fault are successfully detected.

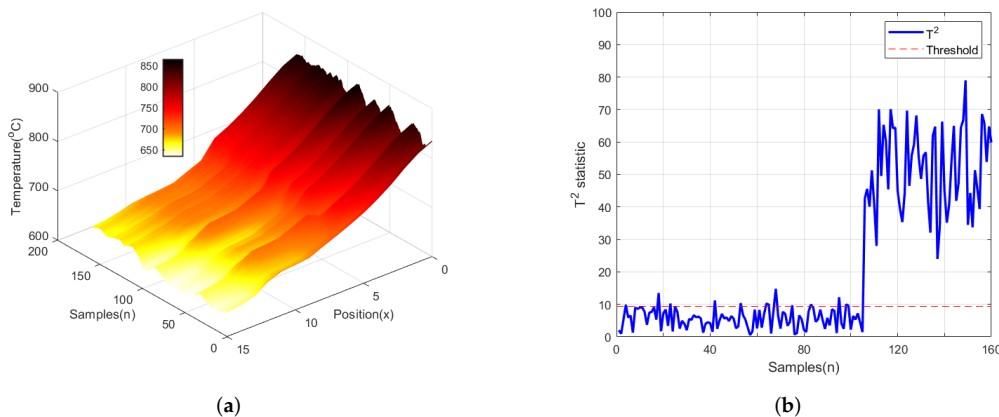

(**a**)  (**b**)

**Figure 12.** The temperature distribution (**a**) and monitoring result (**b**) of fault one.

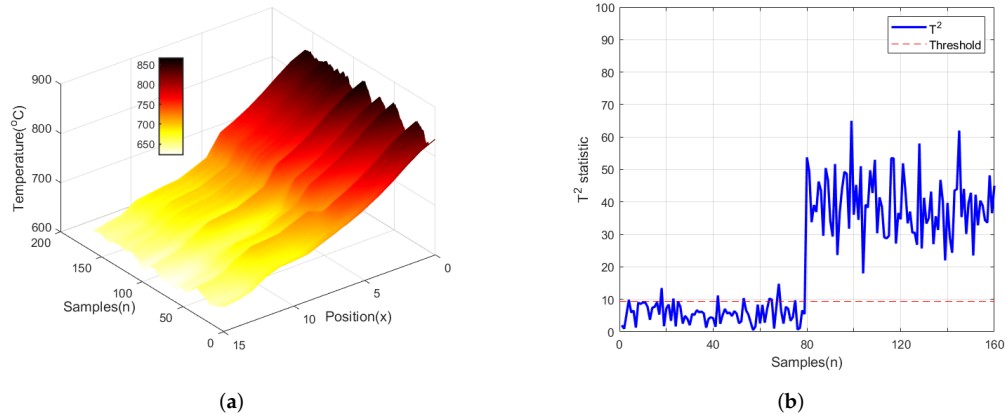

(**a**)  (**b**)

**Figure 13.** The temperature distribution (**a**) and monitoring result (**b**) of fault two.

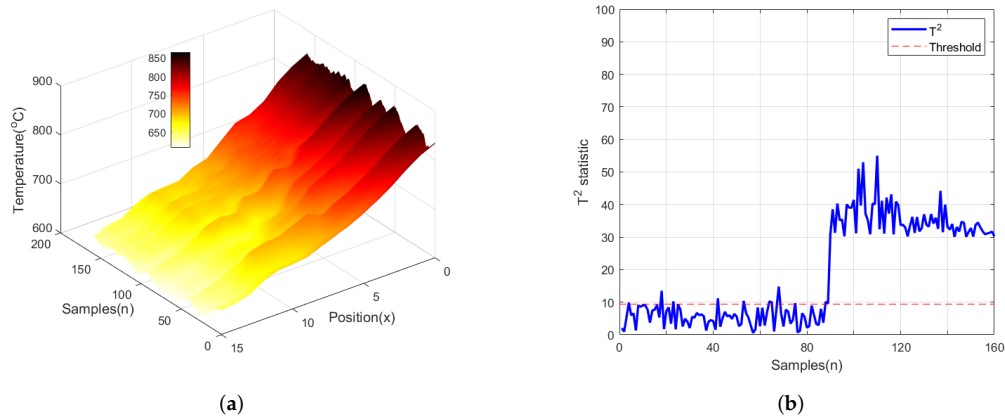

(**a**)  (**b**)

**Figure 14.** The temperature distribution (**a**) and monitoring result (**b**) of fault three.

## 6. Conclusions

In this paper, a modeling and monitoring method with a time–space-coupled nature is proposed for LCP. The global spatio-temporal model is constructed with distributed parameter systems for the LCP and a residual generator is designed to monitor the spatial variation of the strip temperature. The characteristics of the proposed method are as follows:

- The FEM is combined with the Galerkin method to describe the temperature variation in the direction of the length and the thickness, which can accurately monitor the system with a long procedure and meet the requirements of the temperature uniformity in a large spatial range;
- The global spatio-temporal model for the LCP is constructed with the multi-modeling integration method to establish the transition relationship between subsystems. The complexity of modeling is reduced;
- Different types of faults are considered to monitor the faults in the data-driven realization and the effectiveness of the proposed method is verified by actual-process data.

Since the method proposed in this paper can effectively monitor the temperature variation of the strip in each cooling zone, it is difficult to identify the spatial location of the fault and estimate the size of it. In subsequent work, based on the global modeling and monitoring method for the LCP, the research on fault isolation and fault location will be developed and the method will be extended to other distributed parameter models.

**Author Contributions:** Q.W.: conceptualization; writing–original draft preparation; visualization. K.P.: writing–review and editing; visualization; project administration. J.D.: writing–review and editing; supervision. All authors have read and agreed to the published version of the manuscript.

**Funding:** This work was supported by the National Natural Science Foundation of China (NSFC) (U21A20483).

**Institutional Review Board Statement:** Not applicable.

**Informed Consent Statement:** Not applicable.

**Data Availability Statement:** Data and materials will be available upon request.

**Conflicts of Interest:** The authors declare no conflict of interest.

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
