# Peer review of "Modeling and Monitoring for Laminar Cooling Process of Hot Steel Strip Rolling with Time–Space Nature"

_processes, doi:10.3390/pr10030589_

Round 1

Reviewer 1 Report

As a researcher on metal forming, I'd say that authors conduct a good study on modelling and monitoring for laminar cooling process of hot steel strip rolling with time-space nature. There are some useful recommendations for improving the quality of the paper:

1. Some important conclusions could be introduced in the abstract.

2. I suggest that the introduction could be further replenished. Reducing the vibration of the process machine is also a useful method to improve the quality of the metal forming. There is a reference could be cited, which studied the vibration of a metal forming machine and  proposed some original methods to reduce the vibration and improve the quality of the forming parts: Research on vibration model and vibration performance of cold orbital forging machines [J]. Proceedings of the Institution of Mechanical Engineers, Part B: Journal of Engineering Manufacture, 2021, https://doi.org/10.1177/09544054211045011.

3. There are many formulas without sources, and some formula derivation process is not detailed enough.

4. If possible, it is better to conduct a field experiment.

5. Some grammar problems could be corrected.

As a whole, this paper is novel, meaningful and good-quality. It can be accepted after some minor revisions.

Author Response

Response 1:

    Thank you for your helpful suggestions. Due to the fact that the abstract of this paper lacks the description of methods and conclusions for the framework of modelling and monitoring for LCP. According to your valuable suggestions, we have updated the manuscript by adding more measures and conclusions in the abstract.

Response 2:

    Thank you for your helpful and supportive suggestions. Some literatures can be cited for the method of improve the quality of the metal forming. We have updated the introduction and the references with some literatures. 

Response 3:

    Thank you very much for your valuable suggestion. It is our negligence indeed that some the formulas are hard to understand without technical details and sources. It is meaningful to add more technical details for readers to get a better understanding of this paper:

  • In page 4, the source and physical meaning of the boundary conditions are supplied.
  • In page 8, the source of equation (20)-(21) have been supplied, and the derivation process for the global model have been updated.
  • In page 11, the source of equation (28)-(29) and the necessary technical details of equation (37)-(39) for the residual generation based on the kernel representation have been added.
  • In page 12, the descriptions of the threshold setting have been further elaborated. 

Response 4:

    Thanks for your helpful suggestions. The model proposed in this paper for LCP is verified by the process data which are the actual operation ones of a steel plant. We have been to the steel plant to collect data many times. If possible, the experiment about the effectiveness of this model can be conducted in the field.

Response 5:

    Thank you very much for your valuable suggestions. This is our negligence indeed. We have proofread the whole manuscript in all grammar problems and types. We have checked the grammar and spelling throughout the paper, and the grammar and spelling errors have been corrected:

  • In introduction, some descriptions of the literatures have been rephrased.
  • In page 3, the description of LCP has been updated.
  • In page 5-9, the expression of modelling in DPSs have been modified.
  • In page 13-16, the statement about the result in each cooling zone has been rephrased.
  • In page 17, the conclusions of this paper have been arranged.
  • The spelling errors have been corrected of the whole paper.

    Finally, we thank you to timely evaluate our revised paper again, and hope the revision and our response would meet your expectation, and the paper is now suitable for publication in Processes.

Reviewer 2 Report

In this manuscript, the authors present a model for laminar cooling process of hot steel strip rolling and method of monitoring the temperature variation using the combination of FEM and Galerkin method. In general, the approach taken as well as the global model developed have been demonstrated to be viable in describing the spatial variation of the temperature in the LCP of hot steel strips. The analysis and results were explained. The proposed method has also been verified by using the actual process data. The modelling part is acceptable but there are some clarification needed as below:

1) Page 7, referring to the statement “It is necessary to monitor the output of the steel strip in the whole cooling zone … “  this is confusing. What “output” is being referred to?

2) Page 15, confusing statement “The trend of the temperature variation in each cooling zone is fundamental which is similar to that of the predictive CT. So is in the thickness direction.” Please rephrase this sentence.

3) Page 17, the threshold settings in Figures 12, 13 and 14 need to be further elaborated.  

Author Response

Response 1:

    Thank you very much for your concern. “output” in this sentence is referred to the spatial variation of the strip temperature when the steel strip enters from the air cooling zone to the last cooling zone of LCP. Since different working conditions in different zones, the “output” must be monitored by the global model to establish the transition relationship among the cooling zones.

Response 2:

    Thank you for your valuable suggestions. It is our negligence indeed that the descriptions the result in each cooling zone are confusing to understand for readers. This sentence can be replaced by “The trend of the temperature variation in each cooling zone is similar to the prediction of CT, which is also reflected in the temperature gradient in the direction of the thickness.”

Response 3:

    Thank you very much for your helpful comment. It is our negligence that the derivation of the threshold for the residual generator is not elaborated in detail. Please kindly allow us to explain the result of the threshold with the attachment.

    Thank you again for all the helpful and constructive suggestions. We hope the revision and our response will meet your expectation.
